# A Seven-Year Microbiological and Molecular Study of Bacteremias Due to Carbapenemase-Producing *Klebsiella Pneumoniae*: An Interrupted Time-Series Analysis of Changes in the Carbapenemase Gene’s Distribution after Introduction of Ceftazidime/Avibactam

**DOI:** 10.3390/antibiotics11101414

**Published:** 2022-10-14

**Authors:** Matthaios Papadimitriou-Olivgeris, Christina Bartzavali, Eleftherios Karachalias, Anastasia Spiliopoulou, Ekaterini Tsiata, Georgios Siakallis, Stelios F. Assimakopoulos, Fevronia Kolonitsiou, Markos Marangos

**Affiliations:** 1Division of Infectious Diseases, School of Medicine, University of Patras, 26504 Patras, Greece; 2Infectious Diseases Service, Lausanne University Hospital, 1011 Lausanne, Switzerland; 3Department of Microbiology, School of Medicine, University of Patras, 26504 Patras, Greece; 4Independent Researcher, Cardiff CF14 5PQ, UK; 5Department of Pharmacy, University General Hospital of Patras, 26504 Patras, Greece; 6Department of Basic and Clinical Sciences, University of Nicosia Medical School, Nicosia 2408, Cyprus

**Keywords:** KPC, VIM, NDM, bloodstream infection, carbapenemase, carbapenem-resistance, ceftazidime/avibactam, colistin, tigecycline, fosfomycin

## Abstract

Background: Ceftazidime/avibactam (CZA) is a new option for the treatment of KPC-producing *Klebsiella pneumoniae*. The aim of this study was to determine resistance patterns and carbapenemase genes among *K. pneumoniae* (CP-Kp) bacteremic isolates before and after CZA introduction. Methods: *K. pneumoniae* from blood cultures of patients being treated in a Greek university hospital during 2015–21 were included. PCR for *bla*_KPC_, *bla*_VIM_, *bla*_NDM_ and *bla*_OXA-48_ genes was performed. Results: Among 912 *K. pneumoniae* bacteremias: 725 (79.5%) were due to carbapenemase-producing isolates; 488 (67.3%) carried *bla*_KPC_; 108 (14.9%) *bla*_VIM_; 100 (13.8%) *bla*_NDM_; and 29 (4%) carried a combination of *bla*_KPC_, *bla*_VIM_ or *bla*_NDM_. The incidence of CP-Kp bacteremias was 59 per 100,000 patient-days. The incidence of CP-Kp changed from a downward pre-CZA trend to an upward trend in the CZA period (*p* = 0.007). BSIs due to KPC-producing isolates showed a continuous downward trend in the pre-CZA and CZA periods (*p* = 0.067), while BSIs due to isolates carrying *bla*_VIM_ or *bla*_NDM_ changed from a downward trend in the pre-CZA to an upward trend in the CZA period (*p* < 0.001). Conclusions: An abrupt change in the epidemiology of CP-Kp was observed in 2018, due to the re-emergence of VIM-producing isolates after the suppression of KPC-producing ones via the use of CZA.

## 1. Introduction

During the last two decades, carbapenemase-producing Gram-negative bacteria, especially *Klebsiella pneumoniae*, have progressively become prevalent in many countries [1,2]. Carbapenemase-producing *K. pneumoniae* (CP-Kp) isolates have successfully disseminated in most European countries with regional and inter-regional spread, with four countries (Greece, Italy, Malta and Turkey) reaching high levels of endemicity [1].

A variety of carbapenemase genes are prevalent in different regions [1,2,3]. The most common carbapenemase in Europe, and especially in Greece, is *K. pneumoniae* carbapenemase (KPC), belonging to class A [1,4]. Class B—also known as metallo-betalactamases (MBL)—includes, among others, the Verona imipenemase (VIM) and New Delhi metallo-betalactamase (NDM), while class D includes oxacillinases (OXAs), which are mostly prevalent in central Europe [1,4]. In Greece, after the introduction of KPC-producing *K. pneumoniae* in 2007, the prevalent VIM-producing isolates were replaced, and KPC-producing isolates became predominant. Since 2013, NDM-producing *K. pneumoniae* have caused sporadic cases or epidemics, whereas OXA-producing isolates are scarce [1,2].

CP-Kp infections are associated with high morbidity and mortality, mainly due to limited therapeutic options, which for many years included aminoglycosides, colistin, tigecycline, fosfomycin and carbapenems [3,5,6,7]. Nowadays, our armamentarium includes novel b-lactam/b-lactamase inhibitor combinations such as ceftazidime/avibactam; meropenem/vaborbactam; imipenem/cilastatin/relebactam; and the siderophore cephalosporin, cefiderocol [8,9,10]. Since ceftazidime/avibactam was shown to be associated with a better outcome as compared to colistin for the treatment of infections caused by KPC-producing isolates (which predominate in Greek hospitals), it became an important addition for the treatment of the aforementioned infections [10]. This introduction in January 2018 was followed by the replacement of KPC-producing isolates in the ICU by MBL-producing ones over a one-year period [9].

The aim of the present study was to evaluate: the evolution of antimicrobial resistance; carbapenemase gene presence among *K. pneumoniae* bloodstream infections (BSIs) in a Greek university hospital during a five-year period; and especially, the impact of the introduction of ceftazidime/avibactam.

## 2. Results

### 2.1. Antimicrobial Susceptibility and Genotypes

A total of 1932 pairs of blood cultures became positive for *K. pneumoniae,* reflecting 912 episodes of BSIs from 835 patients. Among them, 725 (79.5%) BSIs were due to carbapenemase-producing isolates: 488 (67.3%) carried *bla*_KPC_; 108 (14.9%) *bla*_VIM_; 100 (13.8%) *bla*_NDM_; 14 (1.9%) carried both *bla*_KPC_ and *bla*_VIM_; eight (1.1%) both *bla*_NDM_ and *bla*_VIM_; and seven (1.0%) both *bla*_KPC_ and *bla*_NDM_. 

Most carbapenemase-producing isolates (694; 95.7%) showed high-level resistance (MIC > 8mg/L) to tested carbapenems (imipenem, meropenem). Resistance rates for aztreonam, amikacin, gentamicin, sulfamethoxazol-trimethoprime and ciprofloxacin were 91.1%, 86.5%, 74.1%, 91.4% and 99.6%, respectively. Colistin and tigecycline resistance was observed in 29.7% and 60.3%, respectively. Fosfomycin susceptibility was tested on 443 isolates, of which 138 (31.2%) were resistant. Ceftazidime/avibactam susceptibility was performed among 238 isolates recovered between 2018 and 2021 carrying only *bla*_KPC_; among them 36 (10.9%) were resistant. The distribution of MICs to the aforementioned antimicrobials of CP-Kp is shown in Appendix A.

*K. pneumoniae* isolates were more commonly recovered from patients hospitalized in ICU (321; 35.2%), followed by medical wards (256; 28.1%); surgical wards (151; 16.6%); hematology wards (97; 10.6%), including an allogeneic hematopoietic stem cell transplantation unit; emergency room (60; 6.6%); and pediatric wards (27; 3.0%), including neonatal and pediatric ICU. The distribution of carbapenem-susceptible and resistant *K. pneumoniae* isolates according to hospital department is shown in Appendix A. The presence of CP-Kp was higher in the ICU (95.6%); followed by hematology wards (82.5%); surgical wards (73.5%); medical wards (70.3%); pediatric wards (55.6%); and emergency room (53.3%). 

Among the 591 adult patients who developed a CR-Kp BSI in wards other than the ICU, only 42 (7.1%) were previously hospitalized in the ICU. During the post-intervention period (2018–21), 20 patients previously hospitalized in the ICU developed a CR-Kp BSI: 13 carrying *bla*_KPC_; four *bla*_VIM_; two both *bla*_KPC_ and *bla*_VIM_; and one *bla*_NDM_ gene. 

### 2.2. Interrupted Time-Series Analysis

The overall incidence of *K. pneumoniae* BSIs was 59 per 100,000 patient-days, and stayed stable during the seven-year period. The CP-Kp incidence also remained stable during the study period, accounting for 47 per 100,000 patient-days (Figure 1). The incidence of CP-Kp changed from a downward pre-intervention trend (*r* −0.417) to a sustained increase (*r* 0.131) in the post-intervention period (*p* 0.007) (Appendix A). In the ICU, a significant change in trend was observed (*r* −0.458, *r* −0.045, *p* 0.026), while in non-ICU wards, the CP-Kp incidence showed a stable downward trend during the pre- and post-intervention period (*p* 0.132). 

Among CP-Kp, BSIs due to KPC-producing isolates decreased over time from 88.5% in 2015 to 39.0% in 2021 (Figure 1), reflecting a statistically significant decrease in incidence (*r* −0.964, *p* < 0.001). The change occurred in 2018 with the appearance of isolates carrying *bla*_VIM_ or *bla*_NDM_ alone, or in association with *bla*_KPC_. BSIs due to KPC-producing isolates showed a comparable downward trend in the pre- and post-intervention period (*r* −0.269, *r* −0.554, *p* 0.067); this was associated with an inversed trend of BSIs due to MBL-producing isolates changing from a downward trend (*r −*0.250) in the pre-intervention period, to an upward trend (*r* 0.497) in the post-intervention period (*p* < 0.001). The same inversed trend among MBLs was observed in both ICU (*p* 0.035) and non-ICU wards (*p* 0.050) (Figure 2). 

### 2.3. Colistin, Tigecycline, Fosfomycin, Aminoglycoside and Ceftazidime/Avibactam Resistance 

Resistance rates among CP-Kp of colistin and ceftazidime/avibactam (only for isolates carrying *bla*_KPC_ during 2018–21) remained stable. A significant decrease in resistance rates of fosfomycin (from 53.3% in 2016 to 31.2% in 2021; *r* −0.109, *p* 0.022) and amikacin (from 87.5% in 2015 to 79.2% in 2021; *r* −0.098, *p* 0.008) was observed. On the other hand, gentamicin-resistance rates increased from 57.3% in 2015 to 84.3% in 2021 (*r* 0.211, *p* < 0.001), which was the same for tigecycline rates (from 51.0% in 2015 to 92.2% in 2021; *r* 0.317, *p* < 0.001).

### 2.4. Antibiotic Consumption

Colistin, tigecycline, fosfomycin and ceftazidime/avibactam consumption is shown in Figure 3. Tigecycline was the most commonly used antibiotic, followed by colistin. Fosfomycin was introduced in 2016 and ceftazidime/avibactam in 2018. The administration of fosfomycin (*r* 0.943, *p* 0.001) and tigecycline (*r* 0.773, *p* 0.042) increased during the study period, while the consumption of colistin and ceftazidime/avibactam remained stable. 

### 2.5. Other Pathogens

Overall, 462 *Staphylococcus aureus* (35% by methicillin-resistant strains; MRSA) and 463 *Pseudomonas aeruginosa* BSIs (36% by carbapenem-resistant strains) were detected. The incidence of *S. aureus* BSIs was 30 per 100,000 patient-days (10 for MRSA). *P. aeruginosa* BSIs incidence was 30 per 100,000 patient-days (11 for carbapenem-resistant strains) (Appendix A). The incidence of *S. aureus* BSI increased during the study period (*r* 0.857; *p* 0.014), however, no change in incidence of BSIs by MRSA (*r* 0.036; *p* 0.039), overall (*r* 0.107; *p* 0.819) or carbapenem-resistant *P. aeruginosa* (*r* −0.286; *p* 0.535) was observed.

## 3. Discussion

Our data showed a stability in the incidence of BSIs due to CP-Kp during the study period. The incidence was higher (59 per 100,000 patient-days) than that reported from other Greek hospitals or hospitals where CP-Kp were endemic (9.2–13.5 episodes per 100,000 patient-days) [3,7,11]. As compared to the data from a previous study in our institution during 2005–14, the incidence was higher during the present study (24 per 100,000 patient-days) [2]. 

The mosaic of carbapenemase genes changed in 2018 with the reappearance of *bla*_VIM_-carrying *K. pneumoniae* which had not caused any BSI since 2012 [2]. As previously shown, the introduction of ceftazidime/avibactam probably led to a sudden and significant shift (*p* < 0.001) in the epidemiology of CP-Kp in the ICU from KPC-producing isolates to MBL-producing ones, mainly due to the resurrection of VIM-producing isolates, which had not caused a BSI in the hospital during 2015–17 [9]. While this change in the palette of CP-Kp in non-ICU wards was less pronounced than in the ICU, it remained statistically significant (*p* < 0.001) and was due to the increased rate of NDM-producing isolates and the introduction of VIM-producing ones. The increase in the MBL-producing isolates in non-ICU wards cannot be explained by a prior hospitalization in the adult ICU, since only 7.1% of CP-Kp BSI in non-ICU wards occurred in patients that were previously treated in the ICU. The observed difference between ICU and non-ICU wards might be explained by the higher administration of ceftazidime/avibactam in the ICU (Figure 2) as compared to other departments; such administration was the independent risk factor for BSIs due to MBL-producing isolates, as compared to KPC-producing ones [9]. Increased use of ceftazidime/avibactam in the ICU may be attributed to the fact that according to the National Antimicrobial Susceptibility Testing Committee guidelines, its empiric use in patients with CP-Kp risk factors was preferentially allowed in ICU patients, while in other wards it was restricted to immunocompromised patients or patients with septic shock. Such a change in epidemiology after the introduction of ceftazidime/avibactam was also observed during the same period in Italy due to the dissemination of NDM-producing isolates [12]. In an in vitro model, in the presence of ceftazidime/avibactam, an initial KPC-producing population of ~10^8^ CFU/mL was overthrown by a minor population (10^3^ CFU/mL) of NDM-producing (10^3^ CFU/mL) [13]. The incidence of BSIs by MRSA and carbapenem-resistant *P. aeruginosa* remained stable during the study period (especially after 2018); this probably rules out the possibility of a breach in infection control protocols, which could explain the change in CP-Kp BSIs epidemiology. 

An important difference between the present study and the previous one performed in our hospital during 2005–14 was the expediential increase in infection due to CP-Kp among patients arriving at the emergency department; infections increased from 9% in 2005–14 to 53% in 2015–19 [2]. This is substantially more important than previous studies from different areas (Brazil, China, Demark, Spain and the United States of America) that reported an incidence of rectal colonization of CP-Enterobacterales of 0.08% to 4.2% [14,15,16,17,18]. The main difference is that those studies were performed in countries where the incidence of CP-Kp is significantly lower than in Greece. Recent hospitalization, prior colonization by CP-Kp or prior antibiotic administration were the main risk factors for such colonization [14,15,16,17,18]. Nevertheless, even though no information was collected for the patients included in our study concerning prior hospitalization or antibiotic administration, the extremely high percentage should prompt a thorough examination of risk factors for CP-Kp in patients examined in Greek emergency rooms. 

The extremely high proportion of BSIs due to CP-Kp in patients hospitalized in hematology wards (including patients with autologous or allogeneic stem cell transplantation), could pose severe implications for the outcome of such patients, since the usually proposed empiric treatment of febrile neutropenia does not cover such pathogens [19]. As previously shown, only 38.4% of neutropenic patients with CP-Kp BSI received appropriate empiric therapy, leading to a high 30-day mortality (30.8%) [19]. These findings reinforce the need for local guidelines for the adequate treatment of such patients. 

Before the introduction of novel beta-lactam/beta-lactamase inhibitor combinations and cefiderocol, the only available therapeutic options were aminoglycosides, colistin, fosfomycin and tigecycline [5]. While resistance to colistin steadily increased from 2005 to 2014, it stayed stable during 2015 to 2021, which was the same for colistin consumption [2]. However, gentamicin resistance increased significantly during the two periods, with only 25.9% of isolates in the present study being susceptible to gentamicin. Resistance rates to the aforementioned antibiotics were comparable to previous reports [3,6]. As far as tigecycline was concerned, 60.3% of CP-Kp isolates were resistant, which was significantly higher than the 16% in 2005–14; the explanation was that the susceptibility breakpoint according to EUCAST changed for ≥2 mg/L to ≥1 mg/L for characterization of resistant strains [20,21]. Despite this, tigecycline administration continued to increase, probably due to its use in *Acinetobacter* spp. infections. Due to the high prevalence of KPC-producing isolates among CP-Kp, and the increasing resistance against other antibiotics, the introduction of ceftazidime/avibactam improved the situation by adding a potent option to our limited armamentarium [10]. Among KPC-producing isolates, despite its increasing administration, resistance rates remained low at 10.9%, which was comparable to previous reports [22,23]. 

The study has several limitations. It is a retrospective study performed in a single Greek university hospital, and even though it may reflect the epidemiology in Greek centers, it might not be extrapolated to other countries with a different incidence and epidemiology of CP-Kp. With the exception of colistin, the susceptibility methods used in the present study were not the reference tests. No molecular investigation of resistance to antibiotics other than carbapenemase genes was performed. Clone identification was not performed, although the appearance of isolates with different carbapenemase gene combinations implies that multiple clones disseminated after ceftazidime/avibactam introduction. 

## 4. Materials and Methods

This retrospective study was carried out in the University General Hospital of Patras (UGHP), Greece, a 770-bed teaching hospital, from January 2015 to December 2021. Patients with at least one positive blood culture for *K. pneumoniae* were included in the study. Multiple episodes of bacteremia from the same patient were included if a duration of at least two months had occurred between two episodes. 

### 4.1. Identification and Antimicrobial Susceptibility Testing

*K. pneumoniae* isolates from the blood cultures of patients hospitalized in UGHP were identified by the Vitek 2 Advanced Expert System (bioMérieux, Marcy l’Etoile, France). Antimicrobial susceptibility testing was evaluated by the agar disk diffusion method against imipenem, meropenem, aztreonam, amikacin, gentamicin, sulfamethoxazol-trimethoprime and ciprofloxacin. Among carbapenemase-producing isolates, Minimum Inhibitory Concentrations (MICs) of imipenem, meropenem and tigecycline were determined by Etest (bioMérieux) in all isolates, whereas MIC of colistin was determined by the broth microdilution method. Fosfomycin’s MIC (Etest; bioMérieux) was evaluated after its demand from the treating physician. Ceftazidime/avibactam’s MIC was determined by Etest (bioMérieux) in all isolates carrying only *bla*_KPC_ from January 2018 onwards (this was the month that it became available for administration). Results were interpreted according to the European Committee on Antimicrobial Susceptibility Testing (EUCAST) criteria [24,25]. 

### 4.2. Genotypes of Isolates

The detection of *bla* genes, encoding important carbapenemase types in our area, was performed by PCR, using specific primers for *bla*_VIM_, *bla*_IMP_, *bla*_KPC_, *bla*_NDM_ and *bla*_OXA-48_ according to published protocols [4,26]. 

### 4.3. Antibiotic Consumption

The consumption of colistin, tigecycline, fosfomycin and ceftazidime/avibactam (data from the Department of Pharmacy) was calculated using the DDD per 1,000 patient-days, as described by the WHO Anatomical Therapeutic Chemical (ATC)/DDD [27].

### 4.4. Other Pathogens

In order to assess the effect of other cofounding factors in the evolution of *K. pneumoniae* bacteremia incidence, the yearly incidence of BSIs by *S. aureus* (and MRSA) and *P. aeruginosa* (and carbapenem-resistant strains) was calculated. An extraction from the electronic database of the Microbiology department was performed on bacteremias by the aforementioned bacteria.

### 4.5. Statistical Analysis

SPSS version 23.0 (SPSS, Chicago, IL, USA) software was used for data analysis. The incidence of carbapenemase-producing and non-producing *K. pneumoniae* bloodstream infection was calculated per 100,000 patient-days. Categorical variables were analyzed using the Fisher exact test or the *chi*^2^ test, as appropriate. Trends of carbapenemase-producing and non-producing rates of *K. pneumoniae*, *S. aureus* and *P. aeruginosa* were assessed over the seven years using Spearman’s correlation analysis. A *p* value of <0.05 was considered significant. 

The Python package Prophet was used for forecasting. Ceftazidime/avibactam was introduced in January 2018. A 3-month window (January–March 2018) was allowed as an intermediate/transition period to commence noticing any effects from the introduction of ceftazidime/avibactam, but also to account for the 3-month rolling mean transformation. The forecast after April 2018 is based on the time series before the introduction of ceftazidime/avibactam prior to January 2018. The forecast represents the expected time series from April 2018 onwards, if ceftazidime/avibactam was not introduced. The comparison between the forecast and the actual values are indicative of the impact of the ceftazidime/avibactam on the data. Due to the high variance in the signal (monthly data points), we used a rolling mean with a 3-month rolling window for smoothing.

In order to compare the pre- and post-intervention periods, we used linear regression to fit the data points in each period, respectively. The straight lines show the linear trend in each period whose slopes can be compared. A statistical test between the correlation coefficients of the lines pre- and post-intervention (always excluding the 3-month transition period) can reveal whether their difference is statistically significant using the p-value of the test. In some cases, the forecast reached negative values. Even though this can be valid from a mathematical point of view, in the real world, the signal cannot become negative. Therefore, any negative values in the forecast were imputed with zero values. The grey area represents the 80% confidence interval around the forecast [28].

## 5. Conclusions

In conclusion, *K. pneumoniae* BSIs remained stable during the last five years, with a high rate of infections due to CP-Kp. An abrupt change in the epidemiology of CP-Kp was observed in 2018 in the ICU, followed by a change in other wards in 2019 due to the replacement of KPC-producing isolates with MBL-producing ones, possibly attributed to the introduction of ceftazidime/avibactam in 2018. 

## Figures and Tables

**Figure 1 antibiotics-11-01414-f001:**
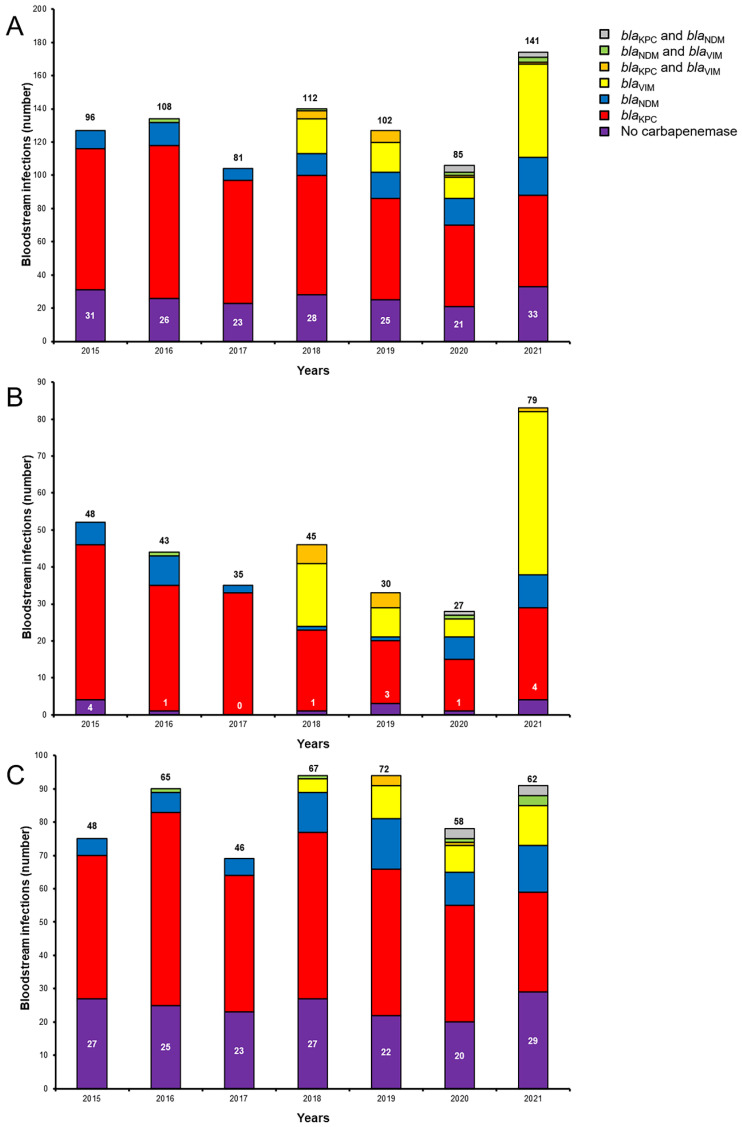
Annual episodes of K. pneumoniae BSIs according to carbapenemase genes; (**A**): all wards; (**B**): ICU; (**C**): non-ICU wards. White numbers: episodes of BSIs due to carbapenem-susceptible *K. pneumoniae*; black numbers: episodes of BSIs due to carbapenem-resistant *K. pneumoniae*.

**Figure 2 antibiotics-11-01414-f002:**
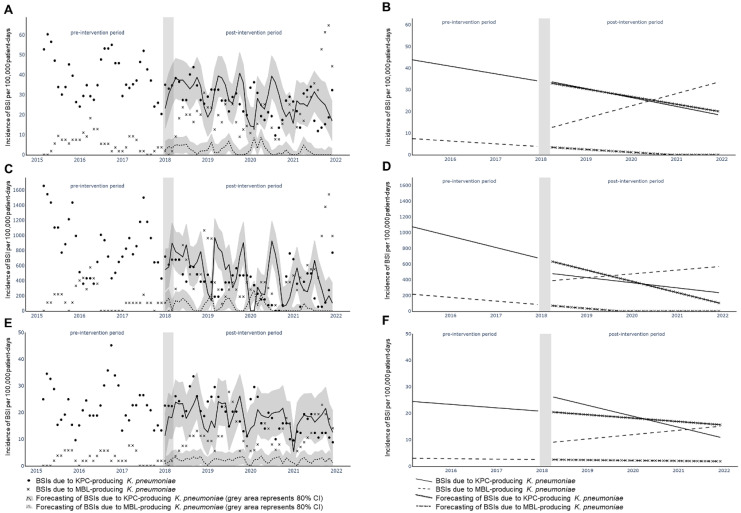
Interrupted time-series analysis of changes in trends of incidence of KPC- and MBL-producing BSIs; (**A**): all wards (actual and predicted values); (**B**): all wards (pre- and post-intervention trends); (**C**): ICU (actual and predicted values); (**D**)**:** ICU (pre- and post-intervention trends); (**E**): non-ICU wards (actual and predicted values); (**F**): non-ICU wards (pre- and post-intervention trends); forecasting trends depict the model predictions by assuming that ceftazidime/avibactam was not introduced on January 2018.

**Figure 3 antibiotics-11-01414-f003:**
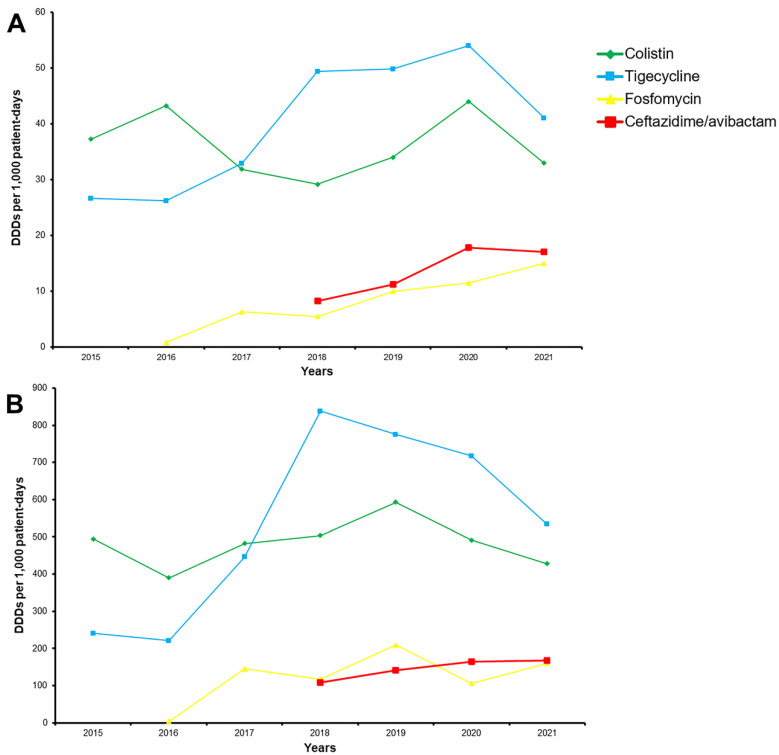
Annual antimicrobial consumption (colistin, tigecycline, fosfomycin, ceftazidime/avibactam) in defined daily doses (DDD) per 1000 patient-days; (**A**): all wards; (**B**): ICU.

## Data Availability

The datasets generated during and/or analyzed during the current study are available from the corresponding author on reasonable request.

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
