# Peer review of "A Seven-Year Microbiological and Molecular Study of Bacteremias Due to Carbapenemase-Producing Klebsiella Pneumoniae: An Interrupted Time-Series Analysis of Changes in the Carbapenemase Gene’s Distribution after Introduction of Ceftazidime/Avibactam"

_antibiotics, 2022, doi:10.3390/antibiotics11101414_

Round 1
Reviewer 1 Report
This single-centre uncontrolled quasi-experimental study used laboratory data from routine clinical cultures and PCR tests performed before and after the introduction of Ceftazidime/avibactam (CZA) in order to investigate temporal associations with the incidence of various types of carbapenemase-producing K. pneumoniae (CP-Kp) bacteraemic isolates.
Overall, the article investigates an important issue. It is mostly well written and appropriately referenced. The methods are generally appropriate, but the statistical methods and models were unclear (thereby difficult to assess their appropriateness). The results are generally well summarised in the text and tables/figures, but some tables appear redundant and improvements are required.
Considering that a randomized experimental design would not be feasible for this investigation, the interrupted time series (ITS) approach provides a good solution. However, a weakness is the absence of a control group or control outcome series (e.g. other pathogen series) and the results should be interpreted with caution accounting for alternative explanations and confounding factors (other than CZA) causing the increase in CP-Kp infections carrying the blaVIM or blaNDM genes.
Please see below for specific comments and suggestions.
Major comments
1. [Lines 258-260] It is unclear why a general presentation of the Prophet package in Python was included. Why the package was used, which statistical model was fitted?
2. [Lines 271 - 278] The use of linear regression is mentioned here, but this does not seem to correspond to the model shown in Figure 2 (which seems more like an ARIMA or GAM model?). Moreover, no information is given regarding how autocorrelation, seasonality and non-stationarity were handled.
3. [Line 279] It is not custom to use a confidence level below 90-95%. Why did you use 80%? (I hope not for making the forecasts/predictions appear more precise!)
4. [Tables 1 and 2] These tables do not offer much in relation to your study objective. Could be deleted or moved to a supplementary file.
5. [Line 109, Suppl Fig 1] There is reference to “Supplementary Figure 1”, which was not provided.
6. [Discussion/interpretation/limitations] The strength of evidence from your investigation would be greatly enhanced if you included in your analysis control data on the occurrence of other pathogens over time (at least one other major gram-negative and one gram-positive organism). I understand this would be a major effort. If not possible, you must acknowledge that you cannot rule out the possibility that confounding factors, other than CZA, such as a breach in infection control measures, an increase in the use of other antibiotics or even a clonal outbreak) may have caused the increase in CP-KPs carrying the blaVIM or blaNDM genes in your hospital. This an important limitation that should be clearly stated, discussed, and incorporated into the interpretation of results.
Minor comments
7. [Line 56]. “Since,” – the comma should be deleted.
8. [Line 68]. “Become” or became?
9. [Figure 2]. You should explain to the readers that the forecasted or predicted values (and lines) refer to the counterfactual scenario assuming CZA was not introduced. You should also explain what the grey coloured areas are.
10. [Figure 2] As currently embedded in the pdf, figure 2 is too blurred and difficult to read. The tif file provided by the authors is high resolution, but it should be more carefully embedded in the manuscript and with larger dimensions and larger font size [I expect this to be sorted during production].
Author Response
Points raised by Reviewer 1
Point 1
[Lines 258-260] It is unclear why a general presentation of the Prophet package in Python was included. Why the package was used, which statistical model was fitted?
Response
A general presentation of the Prophet package in Python was included for the reader as a very short and concise introduction to the basic characteristics of the forecasting tool and method that was used in the paper, given that most researchers in the community are in general not familiar with it. We erased the general presentation of the Prophet package in Python
The package was used to forecast the incidence per 100000 days after the introduction of ceftazidime/avibactam to get an idea of how the time series would have evolved in time if we hadn't introduced it. Even though we create and present at the plots both the forecast (based on data prior to its introduction) and the actual time series after its introduction, eventually we are primarily interested in the linear trend of these two signals. We compare their slopes to get an estimate of the impact of the new treatment.
Prophet can be considered a nonlinear regression model of the form:
y_t = g(t) + s(t) + h(t) + ε_t
where:
g(t) describes a piecewise-linear trend (or “growth term”)
s(t) describes the various seasonal patterns
h(t) captures the holiday effects
ε_t is a white noise error term
Point 2
[Lines 271 - 278] The use of linear regression is mentioned here, but this does not seem to correspond to the model shown in Figure 2 (which seems more like an ARIMA or GAM model?). Moreover, no information is given regarding how autocorrelation, seasonality and non-stationarity were handled.
Response
Prophet is not using an ARIMA model but it is a type of GAM. Linear regression was not used to produce the forecast but to fit a linear trend to the forecast after it was produced by Prophet. The Prophet forecast is basically our approach to getting as close as possible to what the signal would look like if we hadn't introduced ceftazidime/avibactam. Our main focus is not the exact values at each data point of the forecast but the linear trend that the points of the forecast show and how that differs to the linear trend of the data points after the introduction of ceftazidime/avibactam.
- There is a seasonal component in the Prophet model to account for seasonal patterns.
- Prophet does not take autocorrelation on the residuals into account. Since the epsilon term in the additive model represents white noise and therefore Independent and identically distributed random variables, the residual is not assumed to have autocorrelation, unlike ARIMA.
- Stationarity is a requirement to use ARIMA, that is why unit root tests (such as the Dickey-Fuller test) are necessary to ensure the data (or the trend) is stationary and if the null hypothesis is rejected we use differencing until the data is stationary.
However, Prophet’s trend component is always deterministic + possible changepoints and it doesn't assume a stochastic trend unlike ARIMA. Using only deterministic trend underestimates the uncertainty compared to stochastic trend but the use of changepoints and Prophet's future uncertainty estimate compensate for that.
Therefore, testing for stationarity is not a requirement for Prophet.
Point 3
[Line 279] It is not custom to use a confidence level below 90-95%. Why did you use 80%? (I hope not for making the forecasts/predictions appear more precise!)
Response
Even though for hypothesis testing we normally use 90-95% to accept or reject a null hypothesis, for time series forecasting it is custom to use 80% confidence interval (CI) to show the uncertainty around the forecast. As an example, we can refer to the following section of the classic textbook Forecasting: Principles & Practice https://otexts.com/fpp2/prediction-intervals.html. For that reason, 80% is also the default value for the confidence interval calculated in Prophet which is a parameter which we simply left unchanged.
There are two main reasons why we deemed changing this parameter as not necessary:
- Our main goal is to extract the overall trend of the incidence if we hadn't introduced ceftazidime/avibactam so that we can compare with the overall trend from the actual incidence we observed after its introduction. After getting the two (linear) trends we are able to compare them and test whether their slopes are statistically significantly different. So producing the forecast is not about getting exact predictions for each future data point but it's more about capturing the overall trend. If we had used 90-95% for the confidence interval, the grey area of the confidence interval would visually appear as narrower but that wouldn't affect our trend analysis.
- Given the challenges coming from data quality (for example having in some cases very few observations for various variables and points in time or using smoothing to better capture overall trends), it seemed more appropriate to leave the default value for the confidence interval parameter from Prophet unchanged rather than aim for exact predictions with very small uncertainty around them.
Point 4
[Tables 1 and 2] These tables do not offer much in relation to your study objective. Could be deleted or moved to a supplementary file.
Response
We transformed Tables 1 and 2 as Supplementary Tables.
Point 5
Line 109, Suppl Fig 1] There is reference to “Supplementary Figure 1”, which was not provided.
Response
Supplementary Figures and Tables are added
Point 6
[Discussion/interpretation/limitations] The strength of evidence from your investigation would be greatly enhanced if you included in your analysis control data on the occurrence of other pathogens over time (at least one other major gram-negative and one gram-positive organism). I understand this would be a major effort. If not possible, you must acknowledge that you cannot rule out the possibility that confounding factors, other than CZA, such as a breach in infection control measures, an increase in the use of other antibiotics or even a clonal outbreak) may have caused the increase in CP-KPs carrying the blaVIM or blaNDM genes in your hospital. This an important limitation that should be clearly stated, discussed, and incorporated into the interpretation of results.
Response
Yearly incidence of bacteraemias due to P. aeruginosa and S. aureus are provided. While incidence of S. aureus bacteraemia increased during the study period (increase observed even before 2018), no change in the incidence of MRSA and P. aeruginosa (all or carbapenem-resistant) was observed; therefore we can assume that no breaches in infection control practices took place. We added a Supplementary Figure and discussed further this issue in the Discussion.
Point 7
[Line 56]. “Since,” – the comma should be deleted.
Response
Comma is deleted
Point 8
[Line 68]. “Become” or became?
Response
We changed it to became
Point 9
[Figure 2]. You should explain to the readers that the forecasted or predicted values (and lines) refer to the counterfactual scenario assuming CZA was not introduced. You should also explain what the grey coloured areas are. Response
You are right. We better illustrated the point in the methods and the Figures’ description. Grey area around forecasting trends represents 80% confidence interval; it is explained in the Figure legend.
Point 10
[Figure 2] As currently embedded in the pdf, figure 2 is too blurred and difficult to read. The tif file provided by the authors is high resolution, but it should be more carefully embedded in the manuscript and with larger dimensions and larger font size [I expect this to be sorted during production].
Response
I also believe that the production office of the journal will accommodate this issue.
Reviewer 2 Report
The manuscript is well-written and well presented. The statistical analysis is adequate and figures are correct.
Although a molecular characterization of the collection has not been performed, the results provide relevant information regarding the consequences that the clinical use of the ceftazidime-avibactam combination may have had.
However, these results are based on susceptibility tests that are not reference tests. The authors have used agar disk diffusion and gradient strips instead of microdilution to study the antibiotic susceptibilty of the isolates. I am also not very clear about the method used to asses the susceptibility for ceftazidizim-avibactam (this information has to be clear in the manuscript).
Author Response
Points raised by Reviewer 2
Point 1
However, these results are based on susceptibility tests that are not reference tests. The authors have used agar disk diffusion and gradient strips instead of microdilution to study the antibiotic susceptibility of the isolates.
Response
We agree that the susceptibility test used are not reference tests, but concerning the gradient tests (Etests), they have a good correlation to the reference tests. The gradient test was shown to underperform for colistin, thus we used the reference test proposed by EUCAST. Performing microdilution for all the antibiotics tested, is not possible due to shortage of funds and especially understaffing of the microbiology department during the COVID-19 pandemic. These points are discussed in the limitations paragraph.
Point 2
I am also not very clear about the method used to assess the susceptibility for ceftazidime-avibactam (this information has to be clear in the manuscript).
Response
You are right. We added the information in the Materials and Methods.
Round 2
Reviewer 1 Report
The authors addressed all reviewers' comments adequately, resulting in a well presented paper. I congratulate the authors and recommend publication.